# Field Trial of an Automated Batch Chlorinator System at Two Shared Shallow Tubewells among Camps for Forcibly Displaced Myanmar Nationals (FDMN) in Cox’s Bazar, Bangladesh

**DOI:** 10.3390/ijerph182412917

**Published:** 2021-12-08

**Authors:** Nuhu Amin, Mahbubur Rahman, Mahbub-Ul Alam, Abul Kasham Shoab, Md. Kawsar Alome, Maksudul Amin, Tarique Md. Nurul Huda, Leanne Unicomb

**Affiliations:** 1Environmental Interventions Unit, Infectious Diseases Division, icddr,b, 68, Shaheed Tajuddin Ahmed Sarani, Mohakhali, Dhaka 1212, Bangladesh; mahbubr@icddrb.org (M.R.); mahbubalam@icddrb.org (M.-U.A.); akmshoab@icddrb.org (A.K.S.); tarique.huda@icddrb.org (T.M.N.H.); leanne.unicomb60@gmail.com (L.U.); 2Institute for Sustainable Futures, University of Technology Sydney, 235 Jones St., Ultimo, NSW 2007, Australia; 3Action Against Hunger, Dhaka 1212, Bangladesh; washhod@bd-actionagainsthunger.org (M.K.A.); bipu46@gmail.com (M.A.)

**Keywords:** automated chlorine dispenser, underground water chlorination, shallow tubewell, humanitarian, drinking water quality, Bangladesh

## Abstract

Chlorination of shallow tubewell water is challenging due to various iron concentrations. A mixed-method, small-scale before-and-after field trial assessed the accuracy and consistency of an automated chlorinator, Zimba, in Rohingya camp housing, Cox’s Bazar. From August–September 2018, two shallow tubewells (iron concentration = 6.5 mg/L and 1.5 mg/L) were selected and 20 households were randomly enrolled to participate in household surveys and water testing. The field-team tested pre-and post-treated tubewell and household stored water for iron, free and total chlorine, and *E. coli.* A sub-set of households (*n* = 10) also received safe storage containers (5 L jerry cans). Overall mean iron concentrations were 5.8 mg/L in Zimba water, 1.9 mg/L in household storage containers, and 2.8 mg/L in the project-provided safe storage containers. At baseline, 0% samples at source and 60% samples stored in household vessels were contaminated with *E. coli* (mean log_10_ = 0.62 MPN/100 mL). After treatment, all water samples collected from source and project-provided safe storage containers were free from *E. coli*, but 41% of post-treated water stored in the household was contaminated with *E. coli*. *E. coli* concentrations were significantly lower in the project-provided safe storage containers (log_10_ mean difference = 0.92 MPN, 95% CI = 0.59–1.14) compared with baseline and post-treated water stored in household vessels (difference = 0.57 MPN, 95% CI = 0.32–0.83). Zimba is a potential water treatment technology for groundwater extracted through tubewells with different iron concentrations in humanitarian settings.

## 1. Introduction

As of 2015, globally, at least two billion people acquire drinking water from fecal-contaminated sources, responsible for approximately 502,000 deaths each year [1]. Fecal-contaminated water has substantial negative health impacts, particularly in humanitarian and/or emergency settings with crowded spaces and limited access to safe drinking water. Studies conducted during the refugee crisis in the Democratic Republic of Congo in 1994 following an influx of more than 800,000 refugees from Rwanda suggested that water-related pathogens (e.g., *Vibrio cholerae*) were responsible for about 85% of the 50,000 deaths in refugee camps [2,3].

Since 25 August 2017, an estimated 815,280 Myanmar nationals have been forced to flee their native land and settled in Cox’s Bazar, Bangladesh [4,5]. To ensure the availability of drinking water, approximately 20,000 tubewells had been rapidly installed by different organizations in camps housing forcibly displaced Myanmar nationals (FDMN) [6]. Several studies suggest that shallow tubewells can be the source of water-related outbreaks and diseases due to high-level microbial contamination [7,8,9]. A recent report found that of the 14,522 samples from tubewells in FDMN camps, 47% were contaminated with fecal coliforms and 21% with *E. coli* [10]. Contamination of household stored water, or point-of-use (POU) water samples were more common (74% contaminated with fecal coliforms and 35% with *E. coli*) than (source) tubewell water [10]. Despite the progress and efforts made by humanitarian agencies to improve water availability and quality in Rohingya camps, cholera remains a concern [11]. Due to the ongoing threatened health concerns of drinking water with pathogenic microorganisms in refugee camps, there is an essential need for safe drinking water sources to improve water quality.

Point-of-use (POU) chlorination is common in humanitarian settings due to the availability of chlorine products, ease of use, cost-effectiveness, efficacy in inactivating viral and bacterial pathogens, and maintenance of residual chlorine in treated water that protects against recontamination during storage [12]. However, regardless of substantial benefits, several studies revealed that successful chlorination programs in humanitarian settings are challenging owing to several reasons [13,14,15,16,17]. Previous studies from Bangladesh suggested that the taste and smell of chlorine in water, the time required for daily water treatment, personal motivation, knowledge, and behavior change are considerable barriers to POU treatment uptake, resulting in inconsistent and inaccurate practices [14,17,18]. Chlorine-based POU methods usually require optimization of chlorine for a customized dosage dependent on household storage container sizes (i.e., for 5 L, 10 L or 20 L) [19,20].

A recent study conducted to explore treatment efficacy and barriers of drinking water at Rohingya camps, Cox’s Bazar, suggested that although 71% of households in the Camp reported that they used treated water for drinking, only 6.6% of household stored water samples met the WHO recommendation (0.2–2 mg/L) for free residual chlorine levels (mean = 0.09 mg/L). More than 90% of household stored water had lower than recommended (<0.2 mg/L) free residual chlorine [14]. Additionally, centralized chlorinated piped water systems are expensive and require a consistent electricity supply which is not common in humanitarian settings [21].

Few drinking water chlorination technologies are available that overcome some of the challenges of POU water treatment in low-income communities [22,23,24]. In 2012 we conducted a small-scale field trial to explore the accuracy, consistency, and acceptability of an automated chlorinator named Zimba in a low-income urban slum of Dhaka [22]. Zimba, which is attached to the source (i.e., tubewell), automatically treats water with chlorine and was shown to save time and effort and minimized the chance of chlorine dosage errors [22]. Zimba has the practicality of having automatically chlorinated water without the need to install a large-scale system. The Zimba chlorinator is easy to install, and does not require electricity or expert staff to install and operate the system. Compared with centralized chlorination systems, the Zimba chlorinator requires approximately an hour for installation at water points and is a rapid means of accessing automatically chlorinated water. A recent study in Dhaka suggested that Zimba has the potential to overcome some municipal POU water treatment barriers [22]. Zimba disburses a preset dose of three ml of hypochlorite or bleach (NaOCl) solution for every 10 L batch of water into a mixing chamber. Water is flushed immediately after chlorination by a mechanical siphon into a storage tank and dispensed by a tap (Appendix A) [22]. Zimba units do not need specialist engineering for installation and can be adapted to traditional tubewells from which households can draw a small amount of treated water. Despite Zimba’s success in Dhaka, there is a lack of research on how Zimba performs treating water extracted by manual underground shallow tubewell with different iron levels, total dissolved solids (TDS), and water turbidity, and none from humanitarian settings.

To address these research gaps in POU water treatment methods, we conducted a small-scale field trial in two FDMN camps, Cox’s Bazar, Bangladesh, to explore the accuracy, consistency, and acceptability of the Zimba chlorinator in response to demand for safe sources of drinking water.

In this trial, we assessed *E. coli* concentration in untreated tubewell water, pre-intervention water stored in the household vessels, treated Zimba water directly from the Zimba storage tank (immediately after treatment), post-treated water stored in the household vessels, and project-provided safe storage containers at households. We explored the chlorination accuracy and consistency of Zimba, along with community acceptability. We also compared the efficacy of Zimba to deliver adequate residual chlorine in tubewell water with different iron concentrations.

## 2. Materials and Methods

### 2.1. Study Settings and Design

We conducted a mixed-method (quantitative and qualitative) study in two out of thirty Rohingya refugee camps in Ukhiya Upazilas (sub-districts), Cox’s Bazar, Bangladesh (Appendix A), between August and September 2018 (during monsoon season). We used a mixed-method approach to understand possible contradictions between quantitative results and qualitative findings and to reflect participants’ (Zimba users) points of view on acceptability and feasibility of the interventions. These combined methods gave a voice to our study participants and ensured study findings were grounded in participants’ experiences [25]. Both study methods were chosen to assess the feasibility, acceptability, accuracy, and consistency of the Zimba automated chlorinators in Rohingya camps. We randomly selected Camp 1 and Camp 2 from Kutupalong, where the longer-term and recently arrived FDMN population lived together. The FDMN camps in our study area consisted of multiple households with shared toilets, bathing spaces, and drinking water sources owned by the Government of Bangladesh and managed by humanitarian agencies and WASH sector coordinators. Groundwater was extracted through shallow tubewells (<76 m depth) and maintained by Action Against Hunger (ACF). Each tubewell (water collection point) was located within an accessible distance (i.e., within a 5 min walk) for multiple households and was used for drinking and other household purposes. There was no active chlorination project in place when enrolling the study participants. Although there was no active water treatment program in our selected neighborhoods, a small number of households had chlorine tablets (e.g., Aquatab (Medentech, Inc., Wexford, Ireland)) supplied by NGOs.

One of the investigators (NA), along with the field research team, conducted initial scoping visits to the selected camps and performed preliminary water quality assessments measuring iron, turbidity, and total dissolved solids (TDS) using digital field-test kits. We selected four underground shallow tubewells (less than 76 m) from the selected Camps [14] with different water physicochemical properties (tubewell ID-13 and tubewell ID-17 from Camp 1 and tubewell ID-2 and tubewell ID-5 from Camp 2) and based on the following eligibility criteria: (1) water point shared by 90–100 households, (2) water extracted by a manual underground shallow tubewell, (3) the tubewells were the primary drinking water source for the families/users, and (4) had sufficient space around the platform of the tubewell for Zimba installation.

### 2.2. Field Team

Two assigned teams for this study were the field research team and the sample collection team. The field research team conducted surveys and community sessions. The team consisted of experienced staff from the host community who had previous experience in household data collection at the Camp. In addition, they were fluent in Bengali and Rakhine (the mother language spoken by the FDMN). The sample collection team consisted of trained staff who had prior experiences with the water chlorination project, responsible for water sample collection.

### 2.3. Study Participants

The survey team used systematic random sampling to select ten households adjacent to each tubewell to participate in household surveys. Households who reported that they collected water from the intervention tubewells were prioritized for household surveys. Mothers with at least one child less than five years of age were also given enrollment preference for the household surveys.

The survey team visited and provided information on Zimba, described the study activity with household members using cue cards and other visual aids, and obtained their written informed consent in Bengali and Rakhine. They surveyed households at baseline, immediately after Zimba installation of chlorine treatment of the primary tubewells, and at end-line, after four weeks of intervention (Appendix A). After households had been using Zimba, we assessed the acceptability of Zimba and chlorinated water; we conducted in-depth interviews (IDIs) with the study participants (*n* = 10) other than those participating in baseline surveys from among enrolled households. Adult males and females responsible for collecting drinking water were prioritized for IDIs.

### 2.4. Baseline, Seven-Day Follow-Up, and End-Line Data Collection

At baseline, using a structured questionnaire, the survey team conducted household surveys to collect household demography, self-reported drinking water quality, satisfaction and perception, water collection and storage practice, and water treatment methods at households.

Within seven days of Zimba installation, using a structured questionnaire, the survey team conducted follow-up surveys with the study participants to assess immediate acceptability, satisfaction, and problems they were facing with Zimba and chlorinated water. The team tried to address the problems (i.e., increased height of the hand pump, smell/taste of chlorine, and queuing while collecting water) raised by the users in subsequent weeks.

At the end of the 4th week of intervention, the same intervention household members (*n* = 20) were interviewed during the end-line survey. At the end-line, the survey team collected similar information that was collected during baseline and follow-up household visits.

### 2.5. Baseline, Immediate Seven Days and Weekly Water Quality Assessment

At baseline, the sample collection team tested the tubewell and stored water samples from all enrolled households for iron, using an iron meter (Hach color disk test kit, Model IR-18B, Hack Company, Loveland, CO, USA), TDS using a pocket TDS meter (Hanna instruments, Business Park Dr. Vista, CA, USA), turbidity, and free and total chlorine level in collected water using a digital turbidity meter (LaMotte Model 2020i, LaMotte Company, Chestertown, MD, USA) and Colorimeter (LaMotte Model 1200, LaMotte Company, Chestertown, MD, USA). Then, using 100 mL sterile sample collection bags, the sample collection team collected tubewell and water samples stored in all selected households (Nasco Whirl-Pak^®^, 18.5 cm L × 7.5 cm W, 0.064 mm thick, Fort Atkinson, WI, USA). Collected samples maintained at a temperature of <10 °C through primary storage in cold boxes with ice packs were sent to the ACF field laboratory at Cox’s Bazar for *E. coli* contamination assessment.

Within seven days of Zimba installation, the sample collection team performed hourly tests of immediately treated Zimba water every day for eight hours to ensure the source’s chlorine dosing accuracy. During these initial (seven days) follow-ups, the team adjusted the chlorine dosing and addressed the problems (i.e., increased height of the hand pump, smell/taste of chlorine, and queuing while collecting water) raised by the users.

The sample collection team conducted twice a week follow-up visits in the selected households for four weeks. They collected water samples directly from Zimba, and household storage containers after treatment throughout the follow-up visits. We followed similar approaches to test water samples in the field and collect water samples for laboratory testing as described by Amin et al., 2016 [22].

### 2.6. Qualitative Interviews

A total of 10 in-depth interviews (IDIs) were conducted with the study participants other than those participating in baseline surveys from enrolled households (5 from each site) to explore their perception on Zimba and advantages/disadvantages of using Zimba water. One member of the survey team, along with a local Rakhine interpreter conducted IDIs. Each of the interviews focused on Zimba’s user perception, frequency of drinking from Zimba, perceptions (likes/dislikes and advantages/disadvantages/limitations) of treated water, and Zimba chlorination device, changes in taste and/or smell, and perception of treated water over the study period. Study participants were also asked their opinions on how the Zimba device could be improved.

### 2.7. Intervention Components

#### 2.7.1. Chlorine Purchase, Dilution, and Dose Adjustment

The sample collection team purchased household chlorine bleach (5.25% NaOCl: Clorox^®^: the Clorox Company, Oakland, CA, USA) from the market near the Cox’s Bazar field office. They diluted the raw chlorine in tap water (mean iron concentration <0.02 mg/L, mean turbidity 3.7 = Nephelometric Turbidity Unit (NTU) and mean TDS = 10 ppm) to 3.5% NaOCl to attain 2 mg/L of free residual chlorine in household stored water within 24 h of treatment, which they added to Zimba. This process of optimization was checked for each of the tubewells (Appendix A) [26]. The team tested the concentration of chlorine (NaOCl) solution with the colorimeter each time they refilled Zimba. We subsequently reduced the NaOCl concentration to 2.85% for high iron content tubewell water and 2.25% for low iron content tubewell water to attain treated water with 1.5 mg/L of free residual chlorine.

#### 2.7.2. The Zimba Device and Its Installation

The descriptions of Zimba and installation steps were reported previously for the urban Dhaka Zimba trial [22]. In brief, Zimba is made of fiberglass. It has three components:A dispenser made of polypropylene containing diluted bleach (NaOCl);A dosing chamber containing a siphon made of polypropylene;An outer box made of food grade fiberglass that holds the siphon tank and the dispenser.

The production cost of a Zimba chlorinator is estimated at around USD 200, excluding installation, shipment, and cost of chlorine. Zimba does not have any mobile component and does not require any power source. Zimba is mounted on a customized plastic stand approximately 30 cm in height (Appendix A). Two hand-pump mechanics from ACF installed the Zimba under the supervision of icddr,b. We also trained four ACF fieldworkers on regular chlorine refilling and maintenance of Zimba so that it could be used after the intervention period.

#### 2.7.3. Water Storage Containers

During the first follow-up visit, the survey team provided a new 5 L plastic jerrycan (*n* = 10 randomly selected households) of Zimba-chlorinated water (mean residual chlorine at time of distribution = 1.4 mg/L) with an airtight lid as a control. The day before collecting household stored treated water, the field research team refilled the jerrycan and provided it to the respondent’s household. The field research team encouraged the household members to drink water from the jerrycan and to keep the lid closed after use. The field research team also requested the respondents to retain some water (i.e., one glass) in the jerrycan to be collected the next day. This was done to determine if the current household water storage practices impacted the chlorine treatment program.

#### 2.7.4. Intervention Delivery

We used a similar approach for promoting Zimba as the urban Dhaka trial [22]. The day before Zimba installation, the survey team held a meeting with the Camp *Majhi* (local FDMN leader), a local volunteer, and all study participants from the household and introduced water treatment with chlorine and potential health impacts. Zimba usage was explained using flip charts. The survey team requested that the *Majhi* and study participants share and discuss this information with other household members.

### 2.8. Water Sample Laboratory Analysis

After measuring the physiochemical (iron, TDS, water turbidity, and free and total residual chlorine) property of each sample, the sample collection team collected four types of water samples: untreated tubewell water, pre-intervention water stored in the household vessel, treated Zimba water directly from the Zimba storage tank (immediately after treatment), post-treated water stored in the household vessel and project-provided safe storage containers at households. Post-treated water samples were collected at two time points: twice-weekly follow-up visits and during end-line surveys.

All samples were received within 6 h of collection by the laboratory technician and quantified the most probable number (MPN) of *E. coli* per 100 mL of a water sample using the IDEXX- Quanti-tray^®^ 2000 technique with Colilert-24 media (IDEXX Laboratories, Westbrook, Seattle, WA, USA) [27]. In recent water treatment efficacy trials, *E. coli* was most commonly used as an indicator of fecal microbial contamination in drinking water samples [23,28,29]. All water samples were processed on the same day; the laboratory supervisor pre-tested two different dilutions (undiluted (1:1) and 1:10) of water samples to determine the ultimate dilution factor to minimize undetectable samples with *E. coli* or *E. coli* concentrations exceeding the upper detection limit [30]. Due to the low water contamination level found during the initial laboratory analysis, all drinking water samples (both tubewell and household stored water) were analyzed without dilution. We used >1 to <=2419.6 MPN detectable range per tray to detect positive *E. coli* wells within the IDEXX Quanti-Tray [30].

During each day of sample collection, the fieldworkers collected one field blank of distilled water from the ACF laboratory and then tested on the same day with other water samples in the laboratory for *E. coli,* and if we found any growth in the field blank, we reinforced aseptic precautions for subsequent sample collection. The laboratory technician proceeds one laboratory blank per day and one negative control (distilled water) per batch of Colilert per day for quality control. Only 3% (1/30) of the tested blanks and 3% (1/30) negative control had *E. coli* growth. Finally, the laboratory technician processed and sealed the samples in a Quanti-Tray and incubated them at 37 °C for 24 h. The laboratory technician determined the MPN of *E. coli* by counting the number of fluorescing wells and calculating according to the manufacturer’s instructions. We reported all water samples as MPN of *E. coli*/100 mL.

### 2.9. Quantitative Data Analysis

We estimated the value of 0.5 MPN for samples below the detection limit and 2419.6 MPN for samples above the detection limit. We compared *E. coli* contamination and water parameters (iron, TDS, water turbidity, and free and total chlorine) between pre-treated water to post-treated household storage containers, using generalized linear regression models. We also compared levels of iron, free and total chlorine for baseline stored water vs. post-treated household storage containers, baseline stored water vs. post-treated project-provided safe storage container, and post-treated household storage containers vs. post-treated project-provided safe storage container and using a generalized linear regression model, adjusted for clustering at the households with a robust standard error of the mean difference.

### 2.10. Qualitative Data Analysis

The research team and local FDMN interpreters recorded interviews, downloaded, translated, and transcribed them into Bengali, followed by thematic content analysis. The investigator N.A. went through the translations and coded the transcripts according to our research objectives manually. After coding, the investigator thematically categorized the data and matched themes to factors influencing acceptability, feasibility, and chlorine smell and/or taste.

## 3. Results

### 3.1. Baseline Characteristics

#### 3.1.1. Physiochemical Properties of Water from 4 Selected Tubewells

Wide variations of iron concentrations were found in four selected tubewell water (mean iron concentration in tubewell ID-13 was 6.5 mg/L (range: 6.5–6.5 mg/L), 7.5 mg/L (range: 7.5–7.5 mg/L) in tubewell ID-2, 1.5 mg/L (range: 1.5–1.5 mg/L) in tubewell ID-17 and tubewell ID-5). Overall turbidity (mean < 5 NTU) and TDS (mean < 500 ppm) concentrations were low in all four selected tubewell water. TDS concentrations were higher in tubewell ID-2 (TDS = 380 ppm (range: 355–410 ppm)) compared with tubewell ID-17 (TDS = 75 ppm (range: 55–90 ppm)) and tubewell ID-5 (TDS = 155 ppm (range: 135–177 ppm)).

#### 3.1.2. Demographic and Socioeconomic Status

Among the households enrolled in baseline and end-line surveys (*n* = 20), the mean age of the respondents was 37 years, and most (70%) did not have formal education. Each household consisted of six members on average, with at least two < 5-year-old children and an average monthly income of USD 68. Only two households reported no income source for their family, and others were involved in either small business, daily labor, and/or worked at NGOs (Table 1).

#### 3.1.3. Water Collection and Storage Practice

All respondents reported that water was available 24 h a day at their tubewells. Most households (60%) collected their drinking water in a *Kolshi* (traditional metal pot, 5 L and 10 L) and only 30% of stored drinking water was observed fully covered with a solid lid. Almost all households were satisfied with the untreated water supply, including water availability, water quality, and water taste/smell (Table 1). About half of the respondents heard about water treatment “using medicine and/or using tablets,” but only one mentioned the name “chlorine.” One respondent reported a chemical/medicinal smell in their drinking water. At baseline, a few respondents said that they collected drinking water from multiple tubewells if their primary tubewell was broken.

#### 3.1.4. Household Stored Water Quality

At baseline, before installing Zimba, stored water samples in three (15%) households had detectable free residual chlorine within or above the WHO-recommended standard (0.2–2.0 mg/L). However, the mean free residual concentration was low (mean free residual chlorine concentration = 0.15 mg/L, SD = 0.39). In household storage containers, the mean iron concentration was 2.25 mg/L, and 25% of households had high iron concentrations (>5 mg/L). The mean turbidity of water in household storage containers was 2.45 NTU, and the mean TDS was 338 ppm. The concentration of iron in the source tubewell (ID-13) in Camp 1 was 6.5 mg/L and 1.5 mg/L for Camp 2 (tubewell ID-5) (Table 1). All water samples collected directly from tubewells before treatment were free from *E. coli* contamination. At baseline, 12 (60%) water samples in household storage containers were contaminated with *E. coli*, and the mean log_10_ *E. coli* concentration was 0.62 MPN (SD 0.9) (Table 2).

Two out of four installed Zimba chlorinators (installed at tubewell ID-2 and tubewell ID-17) were taken out from the study and were not included in the analysis. Tubewell ID-2 had a very high (7 mg/L) concentration of iron, exerting a chlorine demand which prevented Zimba from providing water with sufficient residual chlorine (≤0.04 mg/L) immediately after treatment over two consecutive days (Appendix A). Additionally, the water turned a red color after treatment due to a chemical reaction between iron and chlorine immediately after mixing. Tubewell ID-17 became too tight to pump water (due to an increase in the height of the barrel of the handpump) after the installation of Zimba. We received complaints from household users that they were unable to pump water. Only three water samples in household storage containers had detectable free residual chlorine (>0.20 mg/L), so we used sample collection bags without sodium thiosulfate tablets during water sample collection [22].

### 3.2. Follow-Up and End-Line Visits

Turbidity and TDS were low and within WHO-recommended acceptable limits [31] in all tubewells and stored water samples. All household stored water samples had <5 NTU turbidity, and more than 60% of water samples had less than 300 ppm TDS. Overall mean iron concentrations were 5.8 mg/L in Zimba water, 1.9 mg/L in household storage containers, and 2.8 mg/L in the project-provided safe storage containers (Table 2).

All post-chlorination Zimba water samples (100%) were within or above the WHO-recommended free residual chlorine level (0.2–2.0 mg/L = 50%, >2 mg/L = 50%) (Mean = 2.1 mg/L, SD = 1.1) and total chlorine (mean = 2.21 mg/L, SD = 0.95). About 94% of post-treated water samples from the project-provided safe storage containers and 24% of post-treated water samples from household storage containers were within or above the WHO-recommended free residual chlorine level. When comparing water characteristics before installing Zimba, we found significant differences in chlorine concentrations between baseline, post-treated water from household storage containers, and project-provided safe storage containers. Mean concentration of free and total chlorine levels at baseline in water from household storage containers were significantly lower (mean difference of free chlorine = −0.23, 95% CI −1.64, 0.17) than from post-treated water in household storage containers. However, mean chlorine levels in post-treated household storage container water samples (taken within 24 h of treatment) were significantly lower ((mean difference of free chlorine = −0.71 (*p* < 0.005)) than the water collected directly from Zimba. The concentration of chlorine was significantly lower in baseline water samples from household storage containers (mean difference of free chlorine = −1.25, 95% CI −1.65, −0.889) and from post-treated water from household storage containers (mean difference of free chlorine = −1.01, 95% CI −1.362, −0.84) compared with the project-provided safe storage containers. The average concentration of free residual chlorine in post-treated Zimba water samples was 2.11 mg/L, post-treated household stored water samples in their vessel was 0.39 mg/L, and post-treated household stored water samples in the project-provided safe storage containers were 1.4 mg/L (Table 2).

All post-treated Zimba water samples (100%) and project-provided safe storage containers were free from *E. coli*. More than 40% of post-treated water samples from household storage containers were contaminated with at least a single *E. coli*. Log_10_ mean *E. coli* concentrations were significantly lower in the project-provided safe storage containers ((log_10_ mean difference = 0.92 MPN, 95% CI 0.59, 1.14)) compared with baseline and at post-treated household storage containers ((log_10_ mean difference = 0.57 MPN, 95% CI 0.32, 0.83)). We did not find a significant difference in *E. coli* concentration between baseline and post-treated water samples from household storage containers ((log_10_ mean difference = 0.34 MPN, 95% CI −0.03, −0.73)) (Table 2).

After treatment, household stored water samples with free residual chlorine within the 0.2–2 mg/L range had less microbial contamination (log_10_-mean *E. coli* = −0.10 MPN/100 mL) compared with samples with chlorine level <0.2 mg/L (log_10_-mean *E. coli* = 0.37 MPN/100 mL; log-mean difference = 0.43, *p* < 0.005). Only 3% of samples were above the detection limit, so this did not affect *E. coli* concentration analysis. A comparison table among WHO guidelines, pre-treated water quality, and post-treated water (Zimba, Jerry can water and treated stored water) is given in the Appendix A.

### 3.3. Acceptability and Perception of Chlorine-Treated Water

Within seven days of Zimba installation, 13 surveyed households (65%) complained about the smell of chlorine/medicine in Zimba-treated water. Only 25% of households were satisfied with the Zimba chlorination system. The main reasons for dissatisfaction were increased height of the tubewell (75%), the long time needed to pump 10 L of water (85%), the smell of chlorine/medicine (70%), and Zimba was not suitable for collecting a small quantity of water (90%). Ten percent of households reported hair loss after bathing in Zimba water (Table 3).

After 4 weeks of Zimba installation, five households (20%) reported that they did not collect water from the Zimba tubewell because they could not tolerate the treated water’s medicinal smell. One household reported that they used to collect Zimba water occasionally. The rest of the 14 intervention households (70%) collected Zimba water regularly (Table 3). After four weeks of intervention, the acceptability of the Zimba chlorinator and/or chlorine smell/taste increased compared with immediately after installation. Only 25% of households complained about the chlorine/medicinal smell in the water, and 80% of households were satisfied with the Zimba chlorinator. Qualitative in-depth interviews suggested that most households (7 out of 10) noticed a strong smell of chlorine at the initial stage after Zimba installation, but they became habituated to the smell after a few weeks. Only 25% of households complained about the increased height of the tubewell, 15% about the time required to pump 10 L water, and 20% about the smell of chlorine/medicine. A greater number of households (75%) expressed that Zimba was not suitable for collecting a small water quantity and 25% of households still complained about hair loss after bathing in Zimba water. One household mentioned the height of the tubewell after Zimba’s installation made it difficult to pump water. After four weeks of intervention, most of the respondents (80%) found the taste/smell of chlorinated water acceptable. They additionally overcame the difficulties faced using Zimba except for collecting small amounts of water (Table 3).

During the in-depth interviews (*n* = 10), all the interviewees clearly understood the Zimba device’s function. They mentioned that Zimba makes the water potable by killing germs. However, all households mentioned that they thought Zimba water was safer for themselves and their children than untreated water. Some households (3 out of 10) mentioned that they had to wait in a long line to collect Zimba water. All the households mentioned that if given proper training on the maintenance and refill of Zimba machines, they would be able to do it themselves; however, they would need technical assistance to repair the machine if broken or nonfunctioning. Only two households mentioned that they would pay a monthly fee to use Zimba. All the other households mentioned that they would have to go back to having untreated, unsafe water if they had to pay for water treatment (Table 3).

### 3.4. Interaction between Iron Concentration and Free Residual Chlorine

The free and total residual chlorine decay rate was higher in tubewell water with a high iron content. The average iron concentration in water from tubewell ID-13 was 6.5 mg/L, and immediately after treatment with 2.85% diluted NaOCl solution, we achieved 1.5 mg/L free residual chlorine. The mean iron concentration in household storage containers near tubewell ID-13 was 3.9 mg/L, we achieved 0.3 mg/L free residual chlorine within 24 h of water collection. The mean concentration of iron in tubewell ID-5 was 1.5 mg/L, and immediately after treatment with 2.25% diluted NaOCl solution, we achieved 2.5 mg/L free residual chlorine. The mean iron concentration in water from household storage containers near the tubewell ID-5 was 0.7 mg/L. We achieved 0.4 mg/L free residual chlorine within 24 h of water collection. There was no difference in free and total residual concentration in household stored water when using a different dilution of NaOCl to treat water (Table 4). For both tubewells, all water samples immediately after treatment had >0.2 mg/L free residual chlorine. However, more than 75% of stored post-treated water from household storage containers had free residual below 0.20 mg/L during follow-up visits (Table 3). The concentration of free residual chlorine was consistently higher during all follow-up visits in the project-provided safe storage containers compared with post-treated household storage containers (Figure 1).

## 4. Discussion

Zimba has the potential to treat groundwater extracted through tubewells with different iron concentrations in an emergency setting. Previously it has only been piloted in urban settings with a stable municipal water supply, and therefore we conducted this trial to understand the treatment efficacy and acceptability in this setting. This was the first-of-its-kind attempt to treat underground shallow tubewell water using an automated chlorinator in a rural and humanitarian setting, albeit on a very small scale. Our results suggested that the concentration of free and total residual chlorine in water samples collected directly from Zimba and the project-provided safe storage containers was consistently observed to be within and above the WHO-recommended range (0.2–2 mg/L). We also found that the project-provided safe storage containers had significantly higher free and total residual chlorine concentrations than baseline and post-treated water from household storage containers. Alam et al., 2020, conducted a study among FDMN camp households that were provided centrally chlorinated water or chlorine tablets to treat water. The study found that less than 13% of household post-treated stored water provided piped water chlorination, and 3% of provided Aquatabs met the WHO-recommended free residual chlorine level [14]. Sikder et al., 2020, found that among FDMN camp households’ stored water, 71% of the bucket, 36% of in-line, and 60% of piped chlorinated water met the WHO-recommended free residual chlorine level [32]. However, our current Zimba chlorination study demonstrated that about 94% of post-treated water samples from the project-provided safe storage containers were within or above the WHO-recommended free residual chlorine level.

The project provided safe storage containers that had significantly lower *E. coli* concentration, higher residual chlorine than baseline, and post-treated water from household storage containers. This suggests that households likely unsafely managed water stored in vessels; at baseline, we observed that 70% of the household storage containers were not appropriately covered. Recently, it was reported that water sources in FDMN camps were unevenly distributed. Many people walk long distances through densely inhabited camps after collecting water [14,21], providing an opportunity for contamination due to unsafe handling during transport. Furthermore, household storage containers in humanitarian settings often had observed fecal contamination likely due to insufficient regular cleaning [33]. The project provided safe storage containers which had a small opening and were likely accessed less frequently [34], providing fewer opportunities for contamination and chlorine dissipation. This signifies that it is vital for successful water chlorination projects in humanitarian settings to ensure that users collect water from treated sources in safe storage containers for recommended periods [35,36]. Safe handling from collection to pouring into a glass for drinking is crucial for a successful water treatment program [37].

Our results suggested that only 24% of post-treated water samples from household storage containers were within or above the WHO-recommended free residual chlorine level. More than 41% of drinking water in our post-treated household storage containers was also found contaminated with at least one MPN *E. coli*/100 mL. Possible explanations of residual chlorine failure and *E. coli* in treated water include collecting water from other tubewells, low or no residual chlorine from Zimba due to inconsistencies in dosing, chlorine demand from the interaction between iron and NaOCl, and/or due to unsafe handling of stored water that leads to recontamination [38]. Recent studies from rural Bangladesh also reported an inverse relationship between free residual chlorine and iron concentration in tubewell water [39], which confirms findings from several studies that the high-water iron content reduces the microbial effectiveness of chlorination [40,41].

Unlike current water chlorination technologies (i.e., Aquatab or bucket chlorination), Zimba has the potential to treat underground water with different turbidity, iron, and TDS concentrations. The dose of chlorine (NaOCl) solution can also be adjusted depending on the physicochemical water quality. Our results suggested that different doses of chlorine (NaOCl or NaDCC) are required to treat groundwater in the FDMN camps. Our study recommends that if the water source concentration of iron exceeds 7.0 mg/L chlorination should be avoided [39], or before chlorination, the water should be treated with potassium permanganate (KMnO_4_) and lime to remove iron [41]. We found that the average free residual chlorine from direct Zimba water samples was more than 2.10 mg/L immediately post-treatment, which decreased to 0.39 mg/L in post-treated household storage containers and 1.4 mg/L in the post-treated project-provided safe storage containers within 24 h of treatment. This result is consistent with a recent study conducted in rural Bangladesh on groundwater chlorination, which suggested that groundwater with iron (>3 mg/L) may have low-level or no residual chlorine after chlorination [39]. Future studies should explore the options for treating groundwater with a high concentration of iron in a low-resource setting.

Strong chlorine smell and the taste was the predominant barrier to using chlorine products in the FDMN camps thus limiting the possibility of increasing chlorine concentration [14,32]. Our qualitative study found that respondents from most households (7 out of 10) detected a strong chlorine smell at the initial stage of Zimba installation, but they reported becoming habituated with the smell after four weeks of intervention. This was similarly found in a recent POU chlorination trial in Dhaka, suggesting that a chlorine dose between 0.1 mg/L and 0.2 mg/L total residual chlorine in the first 1–2 months of operation allowed the users to become accustomed to the taste and smell of chlorine [17]. After four weeks of intervention, the Zimba device’s acceptability (70% collected Zimba water regularly and 80% of households were satisfied with the Zimba chlorinator performance) increased significantly compared with immediately (seven days) after Zimba installation. Unlike the Dhaka Zimba trial, we addressed the problems related to increased tubewell height by constructing a raised footstep (Appendix A), the smell/taste of chlorine by adjusting the chlorine dosage, and periodic behavior change communication messages after seven days of Zimba installation. Despite Zimba acceptability, 75% of households reported that Zimba was not suitable for collecting a small water quantity, for example, the volume needed for anal cleansing after defecation. Zimba attached to the tubewell required pumping 10 L to treat each batch of water which was time-consuming when requiring only a small amount of water. An alternative water reservoir located near the latrine may overcome this limitation.

A Zimba unit that can serve up to 50 families is less expensive than a centralized piped water chlorination system. It does not require household-level distribution of chlorine products (i.e., Aquatabs, Halotabs (Halazone 7.5 mg tablet, Sonear Laboratories Ltd., Motijhil, Dhaka) or similar)), which can be expensive and not always feasible in humanitarian settings. Zimba can be constructed in the local markets and does not need a high-tech manufacturer. The concentrations of NaOCl can be adjusted depending on the water iron concentration. Each Zimba chlorinator requires less than an hour to install at water points and no electricity is required for installation. In our study, all Zimbas were installed by NGO handpump mechanics, and the NGO field staff was trained to dilute NaOCl and refill Zimba. These features are optimal for a rapid roll-out of drinking water chlorination programs in emergency settings, especially during an outbreak of water-borne diseases.

In both the Dhaka Zimba trial [22] and during the current trial in FDMN camps, two Zimbas were rapidly uninstalled. In our current study, we removed one Zimba due to a very high (7 mg/L) concentration of iron in the source groundwater. The other became too tight to pump due to an increase in the height of the handpump barrel after installation. In the Dhaka trial, one Zimba was withdrawn because long lines developed, as it took a long time to pump the water. With the second, the amount of space occupied by the device hindered regular cleaning. This suggests that before applying water chlorination in each setting, it is important to study neighborhood contexts, the number of users per water source, type of water supply (i.e., municipal piped, groundwater), and water chemistry (e.g., iron, TDS, turbidity, chlorine).

Despite the success of the trial, this study had some important limitations. We only evaluated the treatment efficacy of two Zimba chlorinators, which may not be sufficient to see statistical differences between the comparison groups. In addition, the household interview sample size was also small, which may have impacted the utility of statistical tests to determine if the differences between groups (e.g., free and total chlorine, and *E. coli* concentrations) were more extensive than would be expected by chance [23]. No other comparable water treatment technologies were available at the time of this study in FDMN camps to compare the Zimba chlorinator’s effectiveness. Further research should compare the dosing and microbial efficiency of Zimba with inline centralized water treatment technologies [22]. Tubewells that were included in this study were not contaminated with fecal bacteria. Results might differ if the source water (tubewell) is contaminated, as reported for numerous camps [8]. Recent POU water treatment trials using automated chlorinators in urban Dhaka suggested that despite the high fecal contamination in source water, household fecal contamination reduced significantly in stored water after chlorination [22,23]. Considering the high risk of waterborne disease outbreaks in humanitarian settings, a water chlorination program should be considered when there is no apparent contamination in the source water, as post-treatment contamination is common in this setting. The study participants were enrolled from only two FDMN camps in Cox’s Bazar; thus, the acceptability, adoption, NaOCl dosing, and treatment efficacy may be different to the host communities and other camps at Cox’s Bazar. We assessed the acceptability of Zimba and chlorine smell/taste only for four weeks. It is not easy to measure the acceptability and adoption within a short period. Adoption of new technology requires a long-term promotion to change behavior [23]. Studies suggest that an aversion to chlorinated water’s taste or smell may limit the use of chlorine products (NaOCl or NaDCC) [17,22]. Adopting a new automated chlorinator depends on the users’ successful performance and initial acceptance. In contrast, the adoption of NaOCl relies on a successful sustained behavior change at the household level [23].

We measured the iron concentration of tubewells only during the monsoon season. However, a seasonal variation in iron concentration has been observed in shallow groundwater aquifers in Bangladesh [15]. Although we only tested the iron and turbidity of tubewell and household stored water, other groundwater inorganic cations similarly interact with chlorine. Research has demonstrated positive correlations between groundwater iron concentration and other chemicals such as arsenic, manganese, and heavy metals; our iron measurement could be indicative of the presence of a range of chemicals that exert chlorine demand [39,42]. There might be some Natural Organic Matter (NOM) in the groundwater in our study tubewells, which might result in disinfection by-product (DBPs) formation after chlorination [43]. Studies have suggested that there are strong associations between consumption of chlorinated water with high DBP (e.g., trihalomethanes) concentrations and adverse health outcomes, including cancer [44]. Finally, we only tested Zimba at two water points in FDMN camps. Elsewhere in FDMN camps, water points could have different physiochemical and/or mechanical differences, altering Zimba’s performance. Future studies with automated chlorinators should focus on large-scale trials with long-term interventions to adopt with the Zimba device and chlorine taste/smell.

## 5. Conclusions and Public Health Relevance

Continuous increases in global refugee populations increase the demand for water treatment technologies. The humanitarian settings rely on a range of water distribution systems, which are largely unprotected. The ongoing demand for safe water in humanitarian settings opens up new opportunities for developing technology for rapid implementation at scale, in addition to large-scale centralized chlorination and household-level treatment. Zimba is cheap and easy to install, but variation in the groundwater chemistry must be addressed. Community-based (decentralized) water treatment systems have the potential to be more affordable and easier to manage than individual household-level treatment by serving more households [23]. Development of community-based shared automated water treatment systems could lead to increased coverage and sustained use of safe water since they do not require changes in daily behavior by every household, the main barrier to scaling up POU approaches [45,46]. The Zimba device is attached/locked to the tubewell and automatically treats water without the active participation of users [22]. Since household members cannot choose whether to chlorinate, they may be more likely to adjust to the consistently chlorinated water’s smell and taste. The Zimba device is easy to install and requires minimal behavior changes; hence, users quickly adapt to the hardware, and acceptability was high after a short trial period. Further water chlorination research is essential to investigate the causes of concentration of residual chlorine in stored water in households for effective water treatment and impact microbial water contamination and health. The results from this study will provide a guideline to the humanitarian WASH sectors to appropriately chlorinate source water, which ultimately improves water quality in the FDMN camp.

## Figures and Tables

**Figure 1 ijerph-18-12917-f001:**
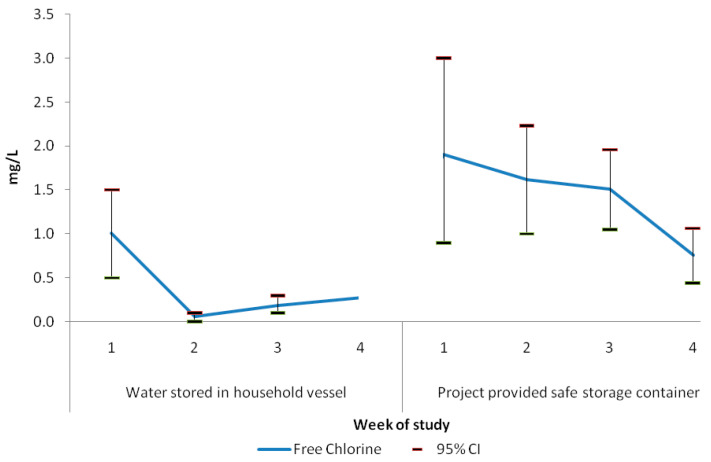
Weekly free residual chlorine in household storage containers (*n* = 80) and project-provided safe storage containers (*n* = 32) during follow-up visits in FDMN camps, Cox’s Bazar, 2018.

**Table 1 ijerph-18-12917-t001:** Demographic characteristics, water collection, storage practice, and drinking water satisfaction level at baseline in FDMN camp, Cox’s Bazar, 2018.

Characteristics	Households *n* = 20, *n* (%)
**Demographics**	
Mean age (years) of the respondents (range)	37 (18–65)
**Education of the respondents**	
No formal schooling	13 (65)
Up to 5th grade	4 (20)
Secondary School	3 (15)
**Education of the household head**	
No formal schooling	7 (35)
Up to 5th grade	5 (25)
Secondary School	6 (30)
College/University	1 (5)
Mean number of children/household <5 (SD)	2 (1.3)
Mean number of household members (SD)	6 (3.2)
**Occupation of the household head**	
Fixed/contractual job	7 (35)
Self-employed or own business	5 (25)
Day labor	6 (30)
Unemployed	2 (10)
^1^ **Household income**	
No income	2 (10)
=<5000 BDT (=< USD 60)	9 (45)
>5000 BDT–<10,000 BDT (>60–< USD 115)	6 (30)
10,000 BDT or above (=> USD 115)	3 (15)
Average household income (BDT) (range)	5705 (0–15,000)
**Water collection and storage practice**	
**Vessels used for drinking water collection**	
*Kolshi* (traditional metal pot, 5 L and 10 L)	12 (60)
Bucket (10 L and 15 L)	4 (20)
Plastic jug (2–3 L)	4 (20)
**Observed Water storage covering**	
Fully covered with solid lid	6 (30)
Partially covered/covered with non-solid lid (paper/perforated lid)	12 (60)
Not covered	2 (10)
**^2^ Level of satisfaction with existing drinking water**	
Generally satisfied with water supply (amount available plus quality)	19 (95)
Water available at sources at a predictable time (Yes)	20 (100)
Water available 24 h	20 (100)
Described water taste as good	19 (95)
Described water taste as not good (e.g., soil, dirt, chemical smell)	1 (5)

^1^ USD = 85 taka. ^2^ Perceived satisfaction reported by the respondents.

**Table 2 ijerph-18-12917-t002:** Water chlorine residual and iron concentration among households during weekly follow up household visits in FDMN camp, Cox’s Bazar, 2018.

Source Water		Baseline Stored Water ^1^	Immediately after Treatment	Post-Treated Follow-up Visits: Stored Water		Mean Differences between Baseline vs. Post-Treated Follow-up Visits		
	Tubewell *n* (%)	Household Own Vessel *n* (%)	Zimba Water *n* (%)	Household Own Vessel *n* (%)	Project-Provided Safe Storage Containers ^3^ *n* (%)	Baseline Household Stored Water Minus Post-Treated Household Own Vessel (95% CI)	Baseline Household Stored Water Minus Project-Provided Safe Storage (95% CI)	Project-Provided Safe storage Container Minus Post-Treated Household Own Vessel (95% CI)
Turbidity (NTU)	*n* = 2	*n* = 20	*n* = 2	*n* = 84	*n* = 15			
<5	2 ^2^ (100)	17 (85)	3 ^2^ (100)	77 (91.7)	14 (93.7)			
5 and above	0	3 (15)	0	7 (8.3)	1 (6.7)			
Mean (SD)	2.55 (2.3)	2.6 (5.7)	2.50 (2.2)	1.8 (3.2)	2.81 (2.7)	0.76 (−1.96, 3.48)	−0.21 (−3.170, 2.749)	0.97 (−2.587, 0.650)
Total Dissolved Solids (ppm)	*n* = 2	*n* = 20	*n* = 2	*n* = 133	*n* = 22			
<300 (excellent)	2 ^2^ (100)	10 (50)	2 ^2^ (66.7)	70 (52.5)	12 (54.6)			
300–500 (fair)	0	10 (50)	1 ^2^ (33.3)	63 (47.3)	10 (45.5)			
Mean (SD)	240.3 (119.9)	238.0 (92.2)	215.3 (115.9)	253.1 (88.9)	235.4 (90.0)	−15.1 (−60.45, 30.29)	2.69 (−54.265, 59.638)	17.76 (−24.588, 60.117)
Iron concentration (mg/L)	*n* = 12	*n* = 20	*n* = 2	*n* = 79	*n* = 17			
<1	0	9 (45)	0	35 (28.2)	10 (58.8)			
1- <3	6 (50.0)	4 (20)	1 (14.3)	27 (34.2)	0			
3- < 5	0	2(10)	0	3 (3.8)	0			
5 or above	6 (50.0)	5 (25)	6 (85.7)	14 (17.7)	7 (41.2)			
Mean (SD)	4.0 (2.1)	2.2 (2.4)	5.8 (1.9)	1.9 (2.1)	2.8 (2.9)	0.59 (−0.846, 1.57)	−0.54 (−2.34, 1.26)	−0.91 (−2.45, 0.63)
Free Chlorine (mg/L)	NA	*n* = 20	*n* = 323	*n* = 133	*n* = 32			
<0.2	NA	17 (85.0)	0	101 (75.3)	2 (6.2)			
0.2–2	NA	3 (15.0)	165 (50.0)	24 (18.0)	22 (68.8)			
>2	NA	0	164 (50.0)	8 (6.0)	8 (25.0)			
Mean (mg/L) (SD)	NA	0.15 (0.4)	2.1 (1.1)	0.39 (0.9)	1.4 (0.9)	−0.23 (−0.64, 0.17)	−1.25 (−1.65, −0.84)	−1.01 (−1.36, −0.67)
Total Chlorine (mg/L)	NA	*n* = 20	*n* = 72	*n* = 133	*n* = 32			
< 0.2	NA	17 (85.0)	0	93 (69.9)	1 (3.1)			
0.2–2	NA	3 (15.0)	31 (43.0)	32 (24.0)	22 (68.8)			
> 2	NA	0	41 (57.0)	8 (6.0)	9 (28.2)			
Mean (mg/L) (SD)	NA	0.17 (0.4)	2.21 (0.95)	0.5 (1.1)	1.72 (1.0)	−0.16 (−0.68, 0.36)	−1.39 (−1.88, −1.90)	−1.23 (−1.66, −0.80)
*E. coli* MPN/100 mL	*n* = 2	*n* = 20	*n* = 8	*n* = 80	*n* = 32			
0	2 (100)	8 (40.0)	8 (100)	47 (58.8)	32 (100)			
1–10	0	6 (30.0)	0	22 (27.5)	0			
> 10	0	6 (30.0)	0	11 (13.7)	0			
Log-mean *E. coli* MPN/100 mL (SD)	−0.30	0.62 (0.9)	−0.30	0.28 (0.73)	−0.30	0.34 (−0.03, 0.73)	0.92 (0.59, 1.24)	0.57 (0.32, 0.83)

^1^ All source water was collected from shallow tubewell. ^2^ We measured turbidity and total dissolved solids only once for two water sources, not for all treated water, considering the low concentration of turbidity and TDS of both selected tube wells. ^3^ A 5 L plastic jerry can.

**Table 3 ijerph-18-12917-t003:** Acceptability and satisfaction with water in households at baseline, within 7 days of Zimba installation, and at end-line in FDMN camps, Cox’s Bazaar, 2018.

Characteristics	Household *n* (%)		
	Baseline *n* = 20	Within 7 Days of Zimba Installation *n* = 20	End-Line (Week 4) *n* = 20
Reported water collection from Zimba			
Always	^2^ NA	NA	14 (70)
Sometimes	NA	NA	1 (5)
Never	NA	NA	5 (20)
Taste of water described by users			
Good	19 (95)	7 (35)	15 (75)
Chemical/chlorine/medicine	1 (5)	13 (65)	5 (25)
Satisfaction with the current water system			
Satisfied	19 (95)	5 (25)	16 (80)
^1^ Reasons for dissatisfaction with the chlorine intervention			
Height of the tubewell increased and it was difficult to pump	NA	15 (75)	5 (25)
Chemical/chlorine/medicine smell	NA	14 (70)	4 (20)
Took a long time to collect water	NA	17 (85)	3 (15)
Tubewell was very hard/stiff to pump	NA	7 (35)	1 (5)
Not suitable for small water collection (one jug/mug)	NA	18 (90)	15 (75)
Difficult for the children and disabled people to collect water	NA	3 (15)	0
Hair loss after bathing in Zimba water	NA	2 (10)	5 (25)
Drinking from this hand pump makes your family members sick			
Often	2 (10)	0	0
Sometimes	3 (15)	1 (5)	1 (5)
Never	14 (70)	18 (90)	18 (90)
Do not know	1 (5)	1 (5)	1 (5)
Do you think drinking directly from this tubewell is safe?			
Yes	4 (20)	19 (95)	16 (80)
Free Chlorine (mg/L)	*n* = 20	*n* = 30	*n* = 20
<0.2	17 (85.0)	16 (53.3)	12 (60.0)
0.2–2	3 (15.0)	7 (23.3)	7 (35.0)
>2	0	7 (23.3)	1 (5.0)
Mean (mg/L) (SD)	0.15 (0.4)	0.75 (1.02)	0.39 (0.57)
Total Chlorine (mg/L)	*n* = 20	*n* = 30	*n* = 20
<0.2	17 (85.0)	14 (46.7)	12 (60.0)
0.2–2	3 (15.0)	9 (30.0)	7 (35.0)
> 2	0	7 (23.3)	1 (5.0)
Mean (mg/L) (SD)	0.17 (0.4)	0.88 (1.12)	0.43 (0.58)
*E. coli* MPN/100 mL	*n* = 20	*n* = 20	*n* = 20
0	8 (40.0)	12 (40.0)	13 (65.0)
1–10	6 (30.0)	5 (16.7)	5 (25.0)
>10	6 (30.0)	13 (43.3)	3 (10.0)
Log-mean *E. coli* MPN/100 mL (SD)	0.62 (0.9)	0.14 (0.6)	0.11 (0.61)

^1^ Multiple responses present. ^2^ NA: Information not collected.

**Table 4 ijerph-18-12917-t004:** Water characteristics for intervention tubewells and household stored water.

Parameters	Tubewell Water Samples		Households Stored Water Samples	
	Tubewell ID-13 *n*	Tubewell ID-5 *n*	Tubewell ID-13 *n* (%)	Tubewell ID-5 *n* (%)
Concentration of iron (mg/L)	*n* = 6	*n* = 6	*n* = 29	*n* = 50
<1	0	0	4 (13.8)	31 (62)
1- <3	0	6	8 (27.6)	19 (38)
3- < 5	0	0	3 (10.3)	0
5 or above	6	0	14 (48.3)	0
Mean (SD)	6.5 (0.4)	1.5 (0.5)	3.9 (2.4)	0.7 (0.3)
Average % of NaOCl used to treat water in Zimba	2.85%	2.25%	2.85%	2.25%
Free Chlorine (mg/L)	*n* = 160	*n* = 177	*n* = 53	*n* = 80
<0.2	0	0	39 (73.6)	62 (77.5)
0.2–2	118 (73.7)	55 (30.1)	13 (24.5)	11 (13.7)
>2	42 (26.2)	122 (68.9)	1 (1.9)	7 (8.7)
Mean (mg/L) (SD)	1.5 (0.9)	2.5 (1.1)	0.3 (0.8)	0.4 (0.9)
Total Chlorine (mg/L)	*n* = 43	*n* = 35	*n* = 53	*n* = 80
<0.2	0	0	36 (67.9)	57 (71.2)
0.2–2	25 (58.1)	12 (34.3)	16 (30.2)	16 (2)
> 2	18 (41.8)	23 (65.7)	1 (1.9)	7 (8.7)
Mean (mg/L) (SD)	1.8 (0.8)	2.4 (1.0)	0.43 (1.1)	0.53 (1.1)

## Data Availability

The data will be shared based on the demand or request from any reader.

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
