# Peer review of "Field Trial of an Automated Batch Chlorinator System at Two Shared Shallow Tubewells among Camps for Forcibly Displaced Myanmar Nationals (FDMN) in Cox’s Bazar, Bangladesh"

_ijerph, 2021, doi:10.3390/ijerph182412917_

Round 1
Reviewer 1 Report
1. If one of the main goals of this study (lines 141-148) was to demonstrate the effectiveness of a tubewell-attached batch chlorinator at disinfecting well water (as understood by many of the participants, lines 514-515), then this effort was a failure. The main reason for this failure was that the coliform content of the source water, as measured by E. coli MPN/100 mL, was zero (Table 2). Hence, the microorganisms in the water exerted negligible chlorine demand, and what little chlorine demand that existed came from the chemical species (geochemical and organic, the latter of which was not measured).
2. The paper works better as a study of perceptions related to chlorination of well water in refugee settlements and camps. The survey work appears to be more valuable than the quantitative work owing to the central flaw mentioned in point no. 1. 3. Data is very poorly presented. For example in Table 2, in addition to the summary statistics presented, frequency numbers in arbitrary concentration ranges are presented for each parameter rather than the values themselves. There is no table in the manuscript or in the Supplementary Information that presents all measured values comprehensively. Such a table should be added to the manuscript or SI. There are other errors in presenting quantitative information; for example in Table 2, standard deviation (SD) is presented even when n=2. It is not clear whether relative percent difference (RPD) substituted SD for n=2. If so, it should be specified. Finally, the only graph presented in the manuscript (Figure 1) is missing a y-axis label. 4. The english usage in the manuscript is adequate; however, the abstract needs to be re-written as it seemed to have been written in haste. In addition, some sections of the manuscript (Sections 3 and 4) are lengthy and verbose as they describe numerical tabulated data in lengthy detail. The paper would read better if such sections are made more concise. 5. Other miscellaneous issues and questions that are not adequately assessed include the following: a. What is the source of the baseline stored water? b. Where are the well details (e.g., presence of sand filter pack at the intake or an open tube like a piezometer; how far down below ground surface is the intake/ well screen)? c. Was the iron measured total or dissolved? If dissolved, was any filtration performed? d. Why were samples collected only in the monsoon season? Were bacterial counts expected to be higher in these months? e. Why is there no discussion of contact time if this is a disinfection study?Author Response
Reviewer #1:
- If one of the main goals of this study (lines 141-148) was to demonstrate the effectiveness of a tubewell-attached batch chlorinator at disinfecting well water (as understood by many of the participants, lines 514-515), then this effort was a failure. The main reason for this failure was that the coliform content of the source water, as measured by E. coli MPN/100 mL, was zero (Table 2). Hence, the microorganisms in the water exerted negligible chlorine demand, and what little chlorine demand that existed came from the chemical species (geochemical and organic, the latter of which was not measured).
- The paper works better as a study of perceptions related to chlorination of well water in refugee settlements and camps. The survey work appears to be more valuable than the quantitative work owing to the central flaw mentioned in point no. 1. 3. Data is very poorly presented. For example in Table 2, in addition to the summary statistics presented, frequency numbers in arbitrary concentration ranges are presented for each parameter rather than the values themselves. There is no table in the manuscript or in the Supplementary Information that presents all measured values comprehensively. Such a table should be added to the manuscript or SI. There are other errors in presenting quantitative information; for example in Table 2, standard deviation (SD) is presented even when n=2. It is not clear whether relative percent difference (RPD) substituted SD for n=2. If so, it should be specified. Finally, the only graph presented in the manuscript (Figure 1) is missing a y-axis label. 4. The english usage in the manuscript is adequate; however, the abstract needs to be re-written as it seemed to have been written in haste. In addition, some sections of the manuscript (Sections 3 and 4) are lengthy and verbose as they describe numerical tabulated data in lengthy detail. The paper would read better if such sections are made more concise. 5. Other miscellaneous issues and questions that are not adequately assessed include the following: a. What is the source of the baseline stored water? b. Where are the well details (e.g., presence of sand filter pack at the intake or an open tube like a piezometer; how far down below ground surface is the intake/ well screen)? c. Was the iron measured total or dissolved? If dissolved, was any filtration performed? d. Why were samples collected only in the monsoon season? Were bacterial counts expected to be higher in these months? e. Why is there no discussion of contact time if this is a disinfection study?
Response: Thank you for your thorough review and critical concern. We have addressed most of your suggestions (but not all), including reviewers #2 and #3. Despite the Zero E. coli in the source water at baseline, this study has an impact on improving drinking water in the camp. Although we agree that there are some critical limitations in this pilot study (we highlighted in the discussion section), we believe based on the experiences in this trial; we will be able to overcome the limitation in the large trials. In addition, if you consider pre-and post-treated free residual in the households and project-provided safe storage containers (Jerry cans), there is a significant improvement in the post-treated water. This is also indicative of successful intervention. Finally, one of the important objectives was to explore the consistent dosing of Zimba, and Zimba successfully performed.
We have revised the abstract based on all reviewer’s suggestions. The contact time of chlorine was a central discussion point in this manuscript. However, the manuscript is already quite long, so we prefer not to include this discussion.

Reviewer 2 Report
In this paper, the authors examined the effect of an automated batch chlorinator system, Zimba, on the water quality in tubewells in Rohingya camps housing. To topic is important as drinking water quality is critical for human health. The iron concentrations and E.coli abundances in the water were investigated in pre- and post-treated water. Overall, the paper is well written, and the content fits well with the aim and scope of IJERPH. I suggest the paper can be published with only a few modifications.
Abstract, please add a sentence talking about the iron concentrations in water with different treatments.
Line 26, "in household" to "in the households"
Line 402, “chlorine”. to “chlorine."
Line 534, "3.9 mg/L, we achieved" to 3.9 mg/L. We achieved"
Line 563, citation format of "Alam et. al., 2020" should be consistent.
Line 633, "Dhaka suggesting" to "Dhaka, suggesting".
The reference format needs to be consistent. Some references use full names, but some references use abbreviations (e.g., 9 and 32). Some have a dot after the abbreviation, but some miss (e.g., 29 and 38).
Author Response
Reviewer #2:
Reviewer 2: Comments and Suggestions for Authors
In this paper, the authors examined the effect of an automated batch chlorinator system, Zimba, on the water quality in tubewells in Rohingya camps housing. To topic is important as drinking water quality is critical for human health. The iron concentrations and E. coli abundances in the water were investigated in pre- and post-treated water. Overall, the paper is well written, and the content fits well with the aim and scope of IJERPH. I suggest the paper can be published with only a few modifications.
Response: Thank you for your nice comments. We have tried to address your concern as following.
Abstract, please add a sentence talking about the iron concentrations in water with different treatments.
Response: We have added a sentence talking about the iron concentrations in water with different treatments in the abstract. Please see changes in page 1, lines 22-24.
Line 26, "in household" to "in the households"
Line 402, “chlorine”. to “chlorine."
Line 534, "3.9 mg/L, we achieved" to 3.9 mg/L. We achieved"
Line 563, citation format of "Alam et. al., 2020" should be consistent.
Line 633, "Dhaka suggesting" to "Dhaka, suggesting".
Response: We have revised the manuscript according to all of your above comments and suggestions.
The reference format needs to be consistent. Some references use full names, but some references use abbreviations (e.g., 9 and 32). Some have a dot after the abbreviation, but some miss (e.g., 29 and 38).
Response: We appreciate your careful review. We have updated all references according to the Int. J. Environ. Res. Public Health author’s instructions.

Reviewer 3 Report
The study "Field Trial of An Automated Batch Chlorinator System 2 at Two Shared Shallow Tubewells among Camps for 3 Forcibly Displaced Myanmar Nationals (FDMN) in 4 Cox’s Bazar, Bangladesh" looks interesting and helpful in attempting to monitor the water quality for the less privileged community. It is good, however, the following improvements might be helpful in further increasing its utility.
1- Abstract:
Please write the abstract clearly. Long sentences such as "A mixed-method, small-scale before-and-after field trial assessed the accuracy and consistency of an automated chlorinator, Zimba in Rohingya camps 16 housing, Cox’s Bazar" and "After treatment, all water samples collected immediately after 24 chlorination at source and project-provided safe storage containers were free from E. 25 coli, but 41% of post-treated water stored in household was contaminated with E. 26 coli. E. coli concentrations were significantly lower in the project-provided safe 27 storage containers [(log10 mean difference=0.92 MPN, 95% CI=0.59-1.14)] compared to 28 baseline and to post-treated water stored in household vessels [(difference=0.57 MPN, 29 95% CI=0.32-0.83)]." could be simplified.
2-Introduction:
The introduction looks a little longer. Please consider shortening it down.
It will be helpful for the reader if the introduction follows a clear structure. Such as (1) general background, (2) problem statement, (3) options available (lit survey), (4) research gap, (5) This study describes.....
The first author might follow the introduction structure of his own published paper "Pickering, Amy J., et al. "Differences in field effectiveness and adoption between a novel automated chlorination system and household manual chlorination of drinking water in Dhaka, Bangladesh: a randomized controlled trial." PloS one 10.3 (2015): e0118397."
3- Materials and Methods
Please establish the scientific basis of "mixed-method" and why it is the most appropriate for this kind of study.
4- Results and Discussion
(1) A comparison table among (1) WHO guidelines, (ii) pre-treated water quality, and (3) post-treated (zimba) water quality will be helpful.
(2) The discussion about iron ions was helpful. Any data on other ionic contaminants such as arsenic?
(3) I assume there might be some NOM in the water as well which might result in disinfectant byproducts. It might be helpful to discuss the tradeoff between chlorination and DBPs in the "Discussion section" where you addressed limitations. This paper might be helpful "Hrudey, Steve E. "Chlorination disinfection by-products, public health risk tradeoffs and me." Water Research 43.8 (2009): 2057-2092."
Conclusions:
It was good to see "The Zimba is cheap and easy to install, but variation in the groundwater chemistry must be addressed".
Author Response
Reviewer #3:
Reviewer 3: Comments and Suggestions for Authors
The study "Field Trial of An Automated Batch Chlorinator System 2 at Two Shared Shallow Tubewells among Camps for 3 Forcibly Displaced Myanmar Nationals (FDMN) in 4 Cox’s Bazar, Bangladesh" looks interesting and helpful in attempting to monitor the water quality for the less privileged community. It is good, however, the following improvements might be helpful in further increasing its utility.
Response: Thank you for your good feedback. We have tried to address your comments as following.
1- Abstract:
Please write the abstract clearly. Long sentences such as "A mixed-method, small-scale before-and-after field trial assessed the accuracy and consistency of an automated chlorinator, Zimba in Rohingya camps 16 housing, Cox’s Bazar" and "After treatment, all water samples collected immediately after 24 chlorination at source and project-provided safe storage containers were free from E. 25 coli, but 41% of post-treated water stored in household was contaminated with E. 26 coli. E. coli concentrations were significantly lower in the project-provided safe 27 storage containers [(log10 mean difference=0.92 MPN, 95% CI=0.59-1.14)] compared to 28 baseline and to post-treated water stored in household vessels [(difference=0.57 MPN, 29 95% CI=0.32-0.83)]." could be simplified.
Response: We have revised the abstract according to your suggestion.
2-Introduction:
The introduction looks a little longer. Please consider shortening it down.
It will be helpful for the reader if the introduction follows a clear structure. Such as (1) general background, (2) problem statement, (3) options available (lit survey), (4) research gap, (5) This study describes.....
The first author might follow the introduction structure of his own published paper "Pickering, Amy J., et al. "Differences in field effectiveness and adoption between a novel automated chlorination system and household manual chlorination of drinking water in Dhaka, Bangladesh: a randomized controlled trial." PloS one 10.3 (2015): e0118397."
Response: We have revised the introduction considering your suggestion.
3- Materials and Methods
Please establish the scientific basis of "mixed-method" and why it is the most appropriate for this kind of study.
Response: We have updated the topics in the respective portion according to your suggestion. Please see changes in page 4, lines 165-169.
4- Results and Discussion
(1) A comparison table among (1) WHO guidelines, (ii) pre-treated water quality, and (3) post-treated (zimba) water quality will be helpful.
Response: Thanks for the suggestion. We have added a comparison table in the supplemental information (tables S2). We have also added text in the result section (See page 12, lines 499-503)
(2) The discussion about iron ions was helpful. Any data on other ionic contaminants such as arsenic?
Response: Unfortunately, due to resource constraints we did not collect information on different ionic contaminants such as arsenic. However, in the discussion section we have highlighted the limitation that we did not measure other groundwater inorganic matter. Please see page 21, lines 716-726.
“Although we only tested the iron and turbidity of tubewell and household stored water, other groundwater inorganic cations similarly interact with chlorine. Research has demonstrated positive correlations between groundwater iron concentration and other chemicals such as arsenic, manganese, and heavy metals; our iron measurement could be indicative of the presence of a range of chemicals that exert chlorine demand [39,42].”
(3) I assume there might be some NOM in the water as well which might result in disinfectant byproducts. It might be helpful to discuss the tradeoff between chlorination and DBPs in the “Discussion section" where you addressed limitations. This paper might be helpful "Hrudey, Steve E. "Chlorination disinfection by-products, public health risk tradeoffs and me." Water Research 43.8 (2009): 2057-2092."
Response: We assume so. We have added the discussion point in the limitation section. Please see the changes in page 21, lines 721-725.
“There might be some Natural Organic Matter (NOM) in the groundwater in our study tubewells, which might result in disinfection by-product (DBPs) formation after chlorination [43]. Studies have suggested that there are strong associations between consumption of chlorinated water with high DBP (e.g., trihalomethanes) concentrations and adverse health outcomes, including cancer”
Conclusions:
It was good to see "The Zimba is cheap and easy to install, but variation in the groundwater chemistry must be addressed".
Response: Thank you for nice feedback.

Round 2
Reviewer 1 Report
I applaud your effort. I hope that you continue to expand this dataset by looking at other sites as well so that you have a more representative and robust datset.